# Smart testing and critical care bed sharing for COVID-19 control

**Paulo J. S. Silva**[1], **Tiago Pereira**[2,3]*, **Claudia Sagastizábal**[1], **Luis Nonato**[2], **Marcelo M. Cordova**[4], **Claudio J. Struchiner**[5]

**1** Instituto de Matemática, Estatística e Computação Científica, Universidade de Campinas, São Paulo, Brazil, **2** Instituto de Ciências Matemáticas e Computação, Universidade de São Paulo, São Paulo, Brazil, **3** Department of Mathematics, Imperial College London, London, United Kingdom, **4** Departamento de Engenharia Elétrica, Universidade Federal de Santa Catarina, Florianópolis, Brazil, **5** Fundação Getúlio Vargas, Rio de Janeiro, Brazil

\* tiago@icmc.usp.br

**Data Availability Statement:** All relevant data, input files, sets of input parameters, and computer code needed to reproduce the computational experiments described in the paper are available in

## Abstract

During the early months of the current COVID-19 pandemic, social distancing measures effectively slowed disease transmission in many countries in Europe and Asia, but the same benefits have not been observed in some developing countries such as Brazil. In part, this is due to a failure to organise systematic testing campaigns at nationwide or even regional levels. To gain effective control of the pandemic, decision-makers in developing countries, particularly those with large populations, must overcome difficulties posed by an unequal distribution of wealth combined with low daily testing capacities. The economic infrastructure of these countries, often concentrated in a few cities, forces workers to travel from commuter cities and rural areas, which induces strong nonlinear effects on disease transmission. In the present study, we develop a smart testing strategy to identify geographic regions where COVID-19 testing could most effectively be deployed to limit further disease transmission. By smart testing we mean the testing protocol that is automatically designed by our optimization platform for a given time period, knowing the available number of tests, the current availability of ICU beds and the initial epidemiological situation. The strategy uses readily available anonymised mobility and demographic data integrated with intensive care unit (ICU) occupancy data and city-specific social distancing measures. Taking into account the heterogeneity of ICU bed occupancy in differing regions and the stages of disease evolution, we use a data-driven study of the Brazilian state of Sao Paulo as an example to show that smart testing strategies can rapidly limit transmission while reducing the need for social distancing measures, even when testing capacity is limited.

## Introduction

Brazil has struggled deeply to curb the transmission of COVID-19. The first case in Brazil was officially reported in late February 2020, after which the number of daily deaths increased rapidly in April and May, plateaued for several months, and then slowly declined in October

the public repository https://github.com/pjssilva/Robot-dance.

**Funding:** TP, PJSS, LN, CS, CJS, MC supported by CEMEAI, the Center for Research in Mathematics Applied to Industry (FAPESP grants 2013/07375-0 and 2015/04451-2), by the Brazilian National Council for Scientific and Technological Development (CNPq; grants 301778/2017-5, 302836/2018-7, 304301/2019- 1, 306090/2019-0, 403679/2020-6) PJSS is supported by FAPESP grant 2018/24293-0. TP is supported by the Royal Society London and by the Serrapilheira Institute (Grant No. Serra-1709-16124).

**Competing interests:** The authors have declared that no competing interests exist.

before resurging in November and December [1–3]. Meanwhile, the number of daily new cases has risen sharply, and more than one year into the crisis, the country has failed to control transmission. A key reason for this failure is the lack of testing strategies and infrastructure. For example, the total stock of RT-PCR test kits for the entire first month of the pandemic, March 2020, was 27,000 for a country with 210 million inhabitants [4]. At a similar stage of the pandemic, roughly the same number of tests were performed daily in Germany, with a population of about 83 million. A similar pattern of daily cases and deaths has been observed in other countries where the availability of intensive care unit (ICU) beds is limited and an efficient and organised testing program has been slow to come into effect [5, 6].

One major unresolved question is whether such a low testing capacity has any utility in helping to curb the spread of COVID-19 and reduce the need for restrictive social distancing measures. Another handicap faced in many countries is the lack of reliable mechanisms for contact tracing. To be effective, tracing needs massive digital data integration as well as measures to ensure the training and safety of personnel [7–11]. The heterogeneous society typical in developing countries adds another layer of complexity. In most countries, the requisite infrastructure is concentrated around hub cities, far from commuter towns [12].

In this situation, nonlinear effects resulting from the high degree of population mobility makes the decision of optimal test distribution a real challenge. Moreover, in such poor countries many individuals cannot isolate as they need to get income [13, 14] and as a result mobility becomes an issue.

In their efforts to control transmission, policymakers struggle to choose where, when, and how many test kits should be distributed when only a limited number is available [15, 16]. In the present report, we describe a strategy based on readily obtainable data to assist decision-makers in this process.

We designed a data-driven smart testing strategy capable of exploring population mobility patterns and ICU bed allocation methods to plan the spatiotemporal distribution of test kits throughout Sao Paulo, a state in Brazil. We integrate anonymised data from mobile devices, census records, and ICU bed usage across multiple areas of the state into an optimisation framework in a complex network of cities, modeling the spread of COVID-19 using an SEIQR compartmental model [17–21]. The approach generates a city-to-city interaction model of transmission and uses testing to alleviate the intensity of social distancing measures and to decrease the pressure on the healthcare system through ICU bed occupancy. Smart testing explores the heterogeneous evolution of viral transmission and can take advantage of mobility patterns in such a way that it efficiently controls spreading in large populations, even when little to no testing is done in population hubs. Indeed, we show that the smart testing strategy is far superior to hub-focused or on-demand testing at reducing the need for mitigation measures.

We provide an analysis for the state of Sao Paulo in Brazil, where all relevant data were collected. Sao Paulo state has a population of 44 million, and as is common in developing countries, its inhabitants are heterogeneously distributed with a major concentration in and around its capital, the city of Sao Paulo. We first assumed that no tests were available and that spreading must be controlled solely by social distancing, thus requiring closure of nonessential services to attain the desired reproduction number. Then, we compared three testing strategies by analysing how each one could help to relax social distancing measures while concurrently reducing the burden on the healthcare system. Smart testing was the superior strategy among those considered. Finally, we analysed a scenario similar to the experienced situation in many countries in which mitigation measures are abandoned after 5 months of control and inhabitants live freely but following sanitary measures. We found that while smart testing alone may be insufficient to completely safeguard the healthcare system, this could be achieved by

introducing a policy of ICU bed sharing between three regions of the state (Sao Paulo city, metropolitan area, and state interior) and by exploiting the different rates at which the disease evolves in these regions. Thus, smart testing succeeds in maintaining an effective healthcare system in a control-free society.

## Results

Assuming that a positive test for COVID-19 alters an individual's behaviour, testing programs affect the mobility patterns of infected individuals who commute between home and work and thus generates nonlinear interactions between regions of the state. Smart testing is capable of exploring the mobility network, heterogeneous stages of disease spreading, and ICU occupancy to plan timely targeted tests in areas that will benefit the whole state. To enable smart testing, we need data on: (i) epidemiological trajectories in each region, (ii) time series of ICU occupancy in each area, (iii) a mobility matrix between regions of the state, (iv) number of daily tests that can be performed, and (v) social distancing criteria. Smart testing then solves an optimisation problem and the output of the algorithm is a spatiotemporal distribution of tests in the state. S1 File provides a flow chart of the required data, and more details on the input data and model are given in the Methods. While the data associated with items (i)-(iv) can be collected easily, the input (v) is flexible and can be tailored in response to specific situations. Mathematically, this translates into a certain objective function that mirrors the measures the government wishes to impose. Here, the objective function study was defined as follows.

The main goal of mitigation measures is to prevent healthcare system collapse by decreasing the value of $R^i(t)$, the effective reproduction number in the $i$th area, for all areas. This can be achieved by applying various mitigation measures [22, 23]. The setting casts $R^i(t)$ as a control variable in an optimal control framework that is then approximated by an optimisation problem. We model each area of the state as an SEIQR model coupled via a mobility matrix (see Methods for details), and a variant of the SEIQR model where the effects of quarantine are described as an effective reproduction number is discussed in S5 File.

With the given data, the platform calibrates a time series to predict the fraction of infected individuals who will require an ICU bed for each day. Based on a probabilistically constrained approach with a confidence level of 90%, maximum ICU bed occupancy remains below the local capacity in all regions. The objective function combines different terms to achieve a balance that provides the most relaxed mitigation measures after the minimisation process. Typical terms are the mean deviation between $R^i(t)$ and $R_0$ (basic reproductive number), a total variation term to avoid too abrupt changes in the control, and terms promoting an alternation of strict measures in nearby cities (see Methods).

To assess the impact of testing strategies, we calibrate the model to the region of interest, as described in S2 File. Here, we provide a full analysis for the state of Sao Paulo, where we gathered all necessary data (see Methods). In S3 File, we also discuss the impact of a smart testing strategy in the early stages of spreading in New York City (NY, USA).

### Smart testing in the state of Sao Paulo

Sao Paulo is the largest and richest state of the Brazilian federation and declared full statewide social distancing measures in late March, 2020. By the end of September, the government-driven testing infrastructure could perform daily COVID-19 RT-PCR tests at the rate of 750 per million inhabitants, which is about 50% of the capacity of most European countries. Thus, whether such a low per capita testing rate can help to reduce transmission is unclear. The healthcare system in Brazil is organised into local health administrative areas composed of

several closely situated cities that share an ICU bed administration system to facilitate allocation. For our study, Sao Paulo state was divided into 22 local health care areas, and we estimated the flow of inhabitants travelling between them by using geolocalised mobile phone data between cities. Combined with demographic information, these datasets enable interactions between areas to be described in the context of COVID-19 spreading. The location of the administrative areas in Sao Paulo state are presented in Fig 1, along with snapshots of the actual epidemic trajectory and actual ICU occupancy on the first day of our study, which spanned approximately 390 days between July 1, 2020 and July 31, 2021. Maps data obtained using geobr, an open-source respository of public domain, official spatial data sets of Brazil [24].

The disease evolution within the state is captured by the epidemiological model in Fig 1. At any given time, inhabitants are considered to be in one of five states: susceptible (green), incubating (pink), infected (red), quarantined (blue), and recovered (grey), and possible transitions between these states are also indicated in Fig 1. We consider asymptomatic implicitly in our

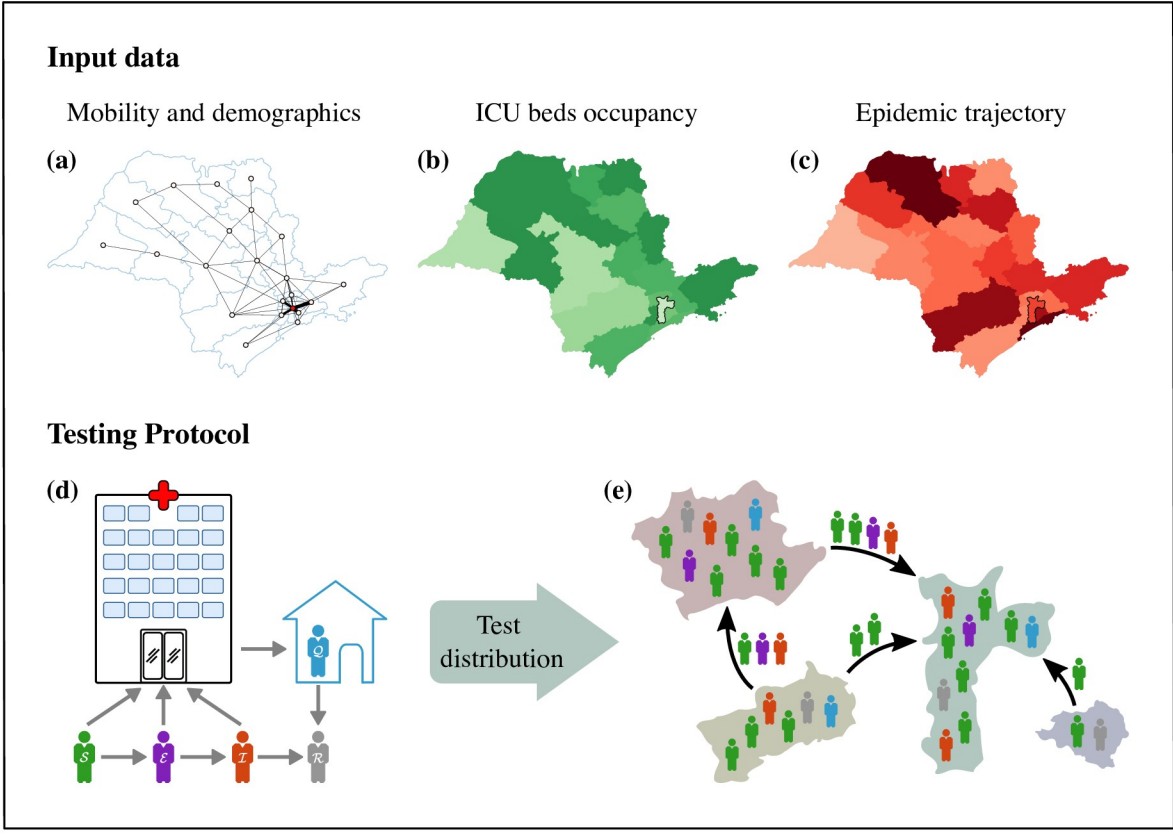

**Fig 1. Smart testing input data for the distribution of COVID-19 test kits for Sao Paulo state.** (a) The geographical outline of Sao Paulo state is shown with a complex network of scattered local hubs created by daily commuting between cities. Edges shown only if more than 2000 people commute between each two hubs. Thicker edges represent more intensive commuting. Sao Paulo city is shown as the red circle. The model integrates the mobility and demographic data shown in (a) together with the dynamics of ICU bed occupancy (b) and the epidemic trajectory (c), for each local health administrative area shown in the maps. The optimisation framework can curb disease spreading by coordinating social distancing measures and test distribution. (d) Within each city, inhabitants are considered to be in one of five states: susceptible ($\mathcal{S}$, green), incubating ($\mathcal{E}$, purple), infected ($\mathcal{I}$, red), quarantined ($\mathcal{Q}$, blue), and recovered ($\mathcal{R}$, grey). If the local health administrative area has been targeted for testing, inhabitants with COVID-19 symptoms who seek healthcare are isolated, tested, and only allowed to leave quarantine if the test result is negative or they regain health. (e) By optimising the distribution of tests together with targeted social distancing measures, we succeed in alleviating the burden on the healthcare system and at the same time allow relaxing of travel restrictions throughout the whole state.

optimisation framework as a restriction on the number of tests. We provide details in Methods. During working hours, the flow of workers commuting into (or out of) the $i$th city can increase or decrease the effective population compared with the city's stable resident population. Other than those in quarantine, we assume individuals continue with their daily commutes between cities. See Methods and references therein for further details.

The model assumes the basic reproductive number with sanitary measures is $R_0 = 1.8$ (the value observed for the second wave [25, 26]). We provide full model details and the parameter values in Methods. S2 File explains the calibration process. We assume that (i) individuals with COVID-19 symptoms seek healthcare and receive the RT-PCR test (if that local health administration is currently testing); (ii) the success detection probability of the test is 80% [27]); (iii) individuals seek healthcare within 3 days after they become infectious ($\tau = 3$); (iv) tested individuals await the test result in isolation and are allowed to leave quarantine only if the test result is negative or they regain health, in which case they join the recovered compartment; and (v) the proportion of false positive tests (uninfected individuals who tested positive) is negligible and has no impact on our results.

Our key proposal is to coordinate social distancing measures with the geographical and temporal distribution of tests. The base case, taken as reference, considers the evolution of the viral transmission rate if no tests are performed and the sole mitigation measure is social distancing [28].

In the absence of tests, we compute $R^i_{\text{no test}}(t)$, the maximum value that the reproduction number attains while keeping ICU occupancy below 90% of the capacity in the ith area. When $R^i_{\text{no test}}(t) = R_0$, no travel restrictions are necessary. This is slightly counter-intuitive but recall that the goal of the protocol is to allow the largest possible $R_t$ that does not lead the failure of the health system. While possible measures to decrease $R_t$ involve travel restrictions when $R_t$ is at its maximum $R_0$ the mitigation measure no longer need to be imposed.

Next, we compute the reproduction number $R^i_{\text{test}}(t)$ of an area under a testing protocol. If testing reduces the need for mitigation measures (i.e., relaxing mobility restrictions without burdening the healthcare system), then

$$R^i_{\text{test}}(t) > R^i_{\text{no test}}(t). \tag{1}$$

When the computed values of $R^i_{\text{test}}(t)$ remain close to $R_0$, this means that testing alone is sufficient to control transmission. However, this is still dependent to a large extent on the history of ICU occupancy and the time between infection and receiving a positive test result. We will address this scenario later by analysing synergy between smart testing and bed sharing.

To compare different strategies, we define the efficiency of a testing protocol as follows. Given the reproduction number $R^i_{\text{test}}(t)$ of the $i$th local health area under a testing protocol at time $t$, improvement (i.e., opening of nonessential services) due to testing can be measured in terms of an increase in the reproduction number, relative to the base case, without testing: $(R^i_{\text{test}}(t) - R^i_{\text{no test}}(t))/R^i_{\text{no test}}(t)$. These values are weighed by the population of the local health area, $N^i$, with respect to the state population, $N^{\text{state}}$, so that the efficiency

$$\mu^i(t) = \left( \frac{R^i_{\text{test}}(t)}{R^i_{\text{no test}}(t)} - 1 \right) \frac{N^i}{N^{\text{state}}} \tag{2}$$

will be positive if testing improves mitigation measures in the $i$th area at time $t$. It might happen that $\mu^i(t) < 0$ for a particular day $t$. However, the goal of the smart testing strategy is to increase the reproduction number of the state as a whole. Therefore, we introduce the efficiency $q$ for the state. Suppose that $K$ represents all regions and that the testing protocol was conducted during $D$ days, totalling $M$ months. The overall efficiency of the testing protocol is

the mean of the individual efficiencies, averaged over the health areas and months:

$$q = \frac{1}{M} \sum_{i \in I} \sum_{t=1}^{D} \mu^i(t) , \qquad (3)$$

where $I$ denotes the set of all health regions. In our calculations, each protocol is applied for $M = 6$ months and $D$ is about 183. Since $\mu^i(t)$ was weighted by the population, the overall efficiency $q$ requires no further normalisation.

We assume that the state of Sao Paulo performs 750 tests per million inhabitants per day. Under this cap we compare three scenarios: (i) smart testing; (ii) testing on demand, that is, the number of tests is proportional to the percentage of infected individuals in the local population; and (iii) testing only in large urban hubs. The last two configurations capture nonlinear effects induced by the concentration of the economic infrastructure in the large cities, which forces a large portion of the population to commute to work daily.

Fig 2 summarises our results. Testing in all three scenarios leads to improvement; however, the efficiency $q$ of testing only in hubs and testing on demand are both 35%, whereas the efficiency of smart testing is vastly superior, at 65%.

## Nonlinear collective effects of smart testing

Testing on demand prioritizes hubs such as Sao Paulo city, where the majority of cases are concentrated (Fig 2d–2f), whereas smart testing exploits the mobility network and finds optimal solutions that require virtually no testing in the hubs (Fig 2a–2c). Instead, the smart strategy focuses on testing key areas while considering ICU occupancy. Remarkably, this strategy leads to an efficiency in the main hub Sao Paulo city as high as when only the hubs are tested. A notable effect of smart testing is that it synergistically generates a collective improvement in the efficiency of testing for the whole state. This results from exploiting the mobility between local health areas and the nonlinear effects emerging from the different stages of the evolution of the disease across the state. It is also striking that the decisions taken with smart testing are not straightforwardly explained by mobility alone.

## Sharing ICU beds and smart testing

We now focus on a pressing societal issue by analysing whether it is possible to reduce or eliminate social distancing control measures, which often leads to the re-emergence of infectious diseases [29]. Here, we aim to control disease spreading after relaxing social distancing by deploying smart testing and coordinated ICU bed sharing alone.

Starting from July 1, 2020, we impose mitigation measures for 140 days and then compare the epidemic trajectories for the subsequent 250 days in two scenarios; one in which all distancing restrictions are removed (while considering sanitary measures), and one in which social distancing restrictions are removed and control occurs through smart testing alone. In the social distancing alone scenario, full capacity of the healthcare system is reached within 30 days and saturation is maintained for nearly 2 months thereafter.

For the smart testing alone scenario, we simulated the number of tests required to control spreading. For $\tau = 3$, testing alone cannot control spreading of the virus without overwhelming the healthcare system. See further details in S4 File.

Next, we consider $\tau = 1$; that is, when infected individuals are identified and isolated 1 day after they become infectious. Note that such early detection of cases would require adoption of new technologies [30]. In this scenario, a daily cap of 5000 tests per million inhabitants suffices to control spreading, as shown in S4 File. This number is about 7-fold higher than the actual

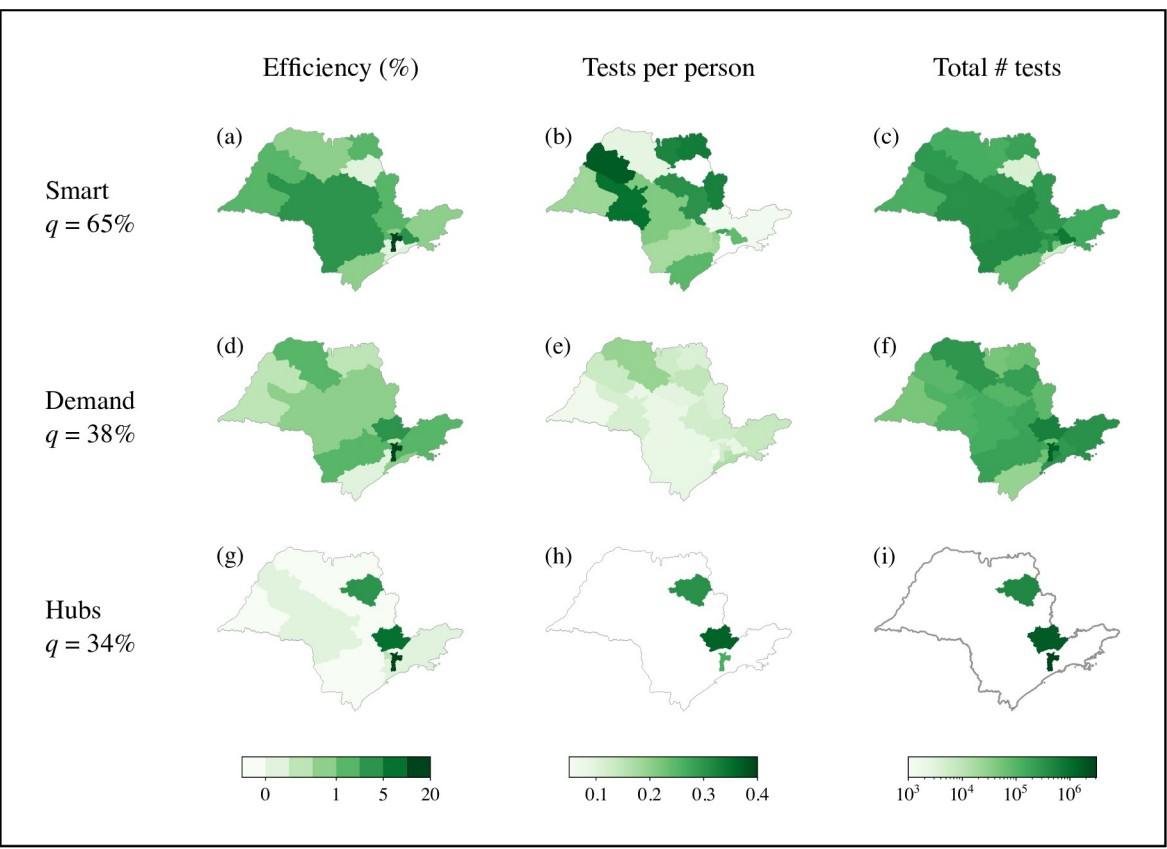

**Fig 2. Effects of testing under different test distribution protocols in the state of Sao Paulo.** The three protocols have a maximum capacity of 750 daily tests per million inhabitants. Sao Paulo city is highlighted with thickened borders in all insets. The panels show the efficiency $\mu^i(t)$ results for local health area (a, d, g), the number of tests per million inhabitants per month (b, e, h), and the total number of tests sent to each area in a 6-month period (c, f, i) for the three protocols: smart testing protocol (a, b, c), on-demand protocol (d, e, f), and hub-only protocol (g, h, i). The numbers on the left report the efficiency $q$ for the entire state. The green colour code indicates efficiency $\mu^i(t)$ in (a, d, g), tests per million in (b, e, h), and total tests performed in each area in (c, f, i) in a period of six months. Note that testing on demand concentrates the tests in Sao Paulo city, while smart testing focuses on the nearby neighbouring regions with little to no testing in Sao Paulo city. Even in this scenario, however, the efficiency in Sao Paulo city is high. Indeed, efficiency is high throughout the state, indicating that exploiting the network structure with testing coordination is beneficial for the entire state.

state testing capacity; nevertheless, this number of tests per day alone would contain disease spreading, thus protecting the healthcare system.

For the state of Sao Paulo, however, there are some nuances that deserve a more detailed analysis. The city of Sao Paulo and its metropolitan area is home to about 50% of the state's population but harbours about 70% of the state's ICU beds. In this situation, pooling and sharing of ICU beds can be effective. Additionally, as in many other countries, the disease evolution in Sao Paulo state has not been uniform. The first cases were detected in Sao Paulo city in late March, almost 2 months before it began to spread to the rest of the state. As a result, by the time the case rate was peaking in rural areas, the rate was stabilising in Sao Paulo city (see more information in S4 File).

Based on the low daily test capacity and the temporal lag between disease emergence in Sao Paulo city and the interior regions, the smart testing strategy would suggest that testing in rural areas can begin later than in the metropolitan area. Our experiments show that, even allowing for the lag in regional transmission, increasing the daily test capacity up to 7-fold

would still be insufficient to prevent the overload of the health system. Thus, a mechanism in which ICU beds are shared between Sao Paulo city and hospitals in the state interior might offer a potential solution. Indeed, we observe a significant improvement in the ICU usage when simulating a partnership between hospitals to share ICU beds for exclusive use by patients with suspected or confirmed COVID-19 infection [31, 32].

The three macro regions of Sao Paulo state to be considered are: Sao Paulo city (population 11.9 million), Sao Paulo metropolitan area (9.4million), and the interior of the state (23 million), which account for 27%, 21%, and 52% of the state's population, respectively (Fig 3a). Under normal conditions, there are 4310 ICU beds in Sao Paulo state, of which 80%, 11%, and 9% are located in Sao Paulo city, the metropolitan area, and the interior, respectively. Considering this unequal distribution of total ICU beds available, we assume that (i) only ICU beds in Sao Paulo city are shared with the other two regions, which reduces the ICU capacity for Sao Paulo city itself; and (ii) the mechanism of ICU bed allocation from Sao Paulo city to the other two regions is based on demographic data. Thus, under the ICU bed sharing agreement, 71.2%, 13.4%, and 15.4% of the total ICU beds are available for Sao Paulo city, the metropolitan area, and the interior, respectively (Fig 3b and 3c).

In this ICU bed sharing scenario, smart testing with a low daily cap of 750 tests per million inhabitants suffices reduce a peak in the ICU usage for the metropolitan area. For the interior area, having shared beds is more relevant than testing, resulting in a 50% increase in the ICU capacity. Thus, combining smart testing and ICU bed sharing to accommodate the heterogeneous

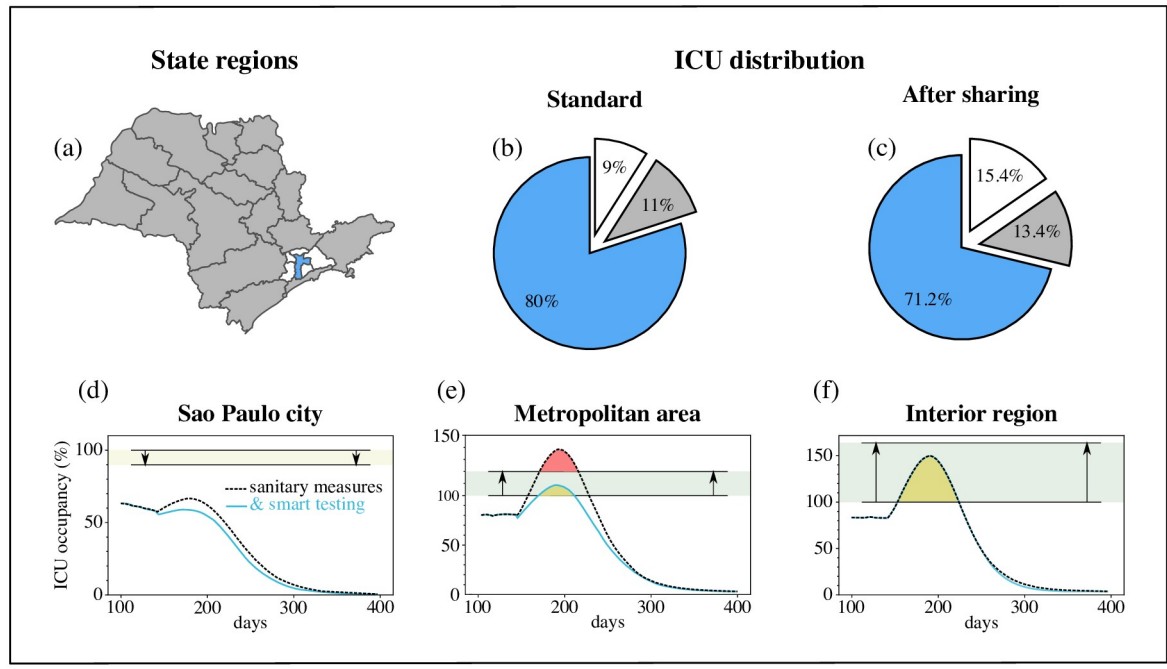

**Fig 3. Sharing of ICU beds and smart testing can control the spread of COVID-19 without restricting travel.** (a) The three macro regions in the state of Sao Paulo are shown: Sao Paulo city (blue), metropolitan area (white), and rural/interior region (grey). (b) The original distribution of the 4310 ICU beds in Sao Paulo state in each of the macro regions. (c) Redistribution of ICU beds after sharing. (d-f) ICU bed occupancy in the three macro regions under the smart testing protocol (blue lines) and without smart testing (black dotted lines). The shaded rectangular areas represent the ICU bed capacity before and after implementation of bed sharing: ~12% loss in Sao Paulo city (d), ~18% gain in the metropolitan area (e), and ~70% gain in the interior (f). The reduction is acceptable for Sao Paulo city, which remains far below maximum local ICU occupancy. The yellow areas show the shared ICU bed use when the testing capacity is 750 daily tests per million inhabitants, and the red areas show the deficit in ICU beds in the metropolitan area when no smart testing is implemented.

infrastructure of the state is not only beneficial to prevent the overload of the healthcare system but can control disease spreading without additional social distancing measures.

We define $\tau$ to be the number of days it takes for an infectious individual to seek assistance. The results reported in Fig 3 were obtained under the premise that susceptible individuals seek healthcare assistance and are tested by RT-PCR on average one day after experiencing symptoms, so that $\tau = 1$. If, in contrast, there is a delay of $\tau = 3$ days between symptom emergence and testing, our simulations show that testing and ICU bed sharing combined will not be sufficient to halt disease spreading. In other words, if $\tau = 3$, restricting travel becomes mandatory. This analysis is in agreement with the efforts being made worldwide to urgently deploy rapid and reliable COVID-19 tests [33].

## Discussion

Control of COVID-19 transmission in low- middle-income societies has been hampered by a scarcity of key resources, including ICU beds, RT-PCR kits, and qualified testing facilities. Although the imminent arrival of effective SARS-Cov-2 vaccines is expected to reduce transmission, it is unlikely that they will be distributed worldwide before the end of 2021. In the meantime, vaccination distribution strategies will initially focus on reducing morbidity and mortality of key subpopulations (healthcare workers and older adults) while maintaining the most critical essential services (healthcare workers engaged in vaccine delivery, teachers, and school staff). Assuming these expectations are satisfied, smart testing strategies of vulnerable populations remain our best hope to curb COVID-19 transmission and preserve the economic health of these societies.

The World Health Organization first called for massive COVID-19 testing in March 2020, and in line with this, we analysed the effect of smart testing in the state of Sao Paulo, Brazil. Our study explored how smart testing can assist in reducing transmission despite challenges due to a low daily testing capacity, unequal healthcare infrastructure, and the evolution of COVID-19 in different areas of the state. Under these adverse conditions, it is critical that we determine how best to handle test distribution in terms not only of where to send kits but also how many and when. To be most effective, these scarce testing resources must be properly deployed. As described in this study, we proposed a smart testing strategy based on mathematical optimisation of test distribution and compared its effectiveness with that of other mitigation protocols.

In all of the configurations considered, the contribution of testing to control of COVID-19 was significant. Interestingly, the smart testing option was far superior to the on-demand testing and hub-only testing scenarios, reaching an efficiency of 65% compared with ~35%. This level of improvement was obtained by considering travel patterns, the state infrastructure, local heterogeneities in disease spreading, and stress imposed on different regional healthcare systems. Our analysis highlights the importance of tests that can deliver results rapidly and also reveals that, even with rapid and reliable testing, not all surges in infection could be controlled by increasing the number of daily tests alone, and that an ICU bed sharing policy may be necessary, especially if daily testing capacity is low. The results and conclusions of this study are robust across models.

By employing data integration and optimisation over a complex network, smart testing provides a striking improvement in the quality of transmission mitigation measures, thereby allowing populations to more rapidly return to pre-pandemic activities. The methodology is applicable to any country facing similar challenges of limited testing capabilities and a highly mobile workforce. The approach can also be easily adapted to accommodate new waves of spreading, and can incorporate more precise medical data on the mechanism of COVID-19 transmission as that becomes available.

Asymptomatic infection is taken into account implicitly. In order to counter any bias that could lead to overestimation of the testing effectiveness, our model combines two features. First, we establish a cap on the proportion of infected individuals that can be detected by testing. Second, the infectivity of the $\mathcal{I}$ compartment should be interpreted as the weighted mean of the infectivity of asymptomatic and symptomatic infected individuals where the weight is given by the relative size of these epidemiological categories. Oran and Topol [34] suggest that at least one third of SARS-CoV-2 infections are asymptomatic. The relative infectivity between symptomatic and asymptomatic COVID-19 infections, when adjusted for age, gender, and serology of the index case, is estimated to be 3.85 in [35]. The effectiveness of any intervention triggered by symptom-based isolation in preventing disease propagation depends on the relative contribution of the presymptomatic and asymptomatic phases to total transmission [36]. The percentage of infections that never developed clinical symptoms (truly asymptomatic) is around 35.0% [37]. Estimates may reach 43% when considering the number of cases that exhibited no symptoms at the time of testing, a group comprised of both asymptomatic and presymptomatic infections. Although we have not accounted for these contingents in our approach, the impact of the smart testing strategies discussed in our work could have been more significant otherwise.

In countries such as Brazil, where control of the pandemic is in the hands of local state and city administrations rather than of the federal government, some relatively affluent cities invested heavily in testing as a means to curb local transmission. This effort was in vain, however, and our analysis using smart testing delivers a strong message; namely, it is better for all citizens if resources are shared with neighbours, and efforts to reduce transmission at individual and local levels are not nearly as effective as collective strategies that concomitantly address all regions. The superiority of smart testing is clear in multiple scenarios, such as financial investment in test kits, stress on the healthcare system, and stress on the population due to restrictive social distancing measures and travel limitations. It is notable that targeted and orchestrated solutions such as smart testing represent a sharp contrast to the local-level solutions currently being implemented in many countries.

## Methods

### An SEIQR model with mobility

We use the notation in [28]. Epidemiological and other relevant parameters are presented in Table 1.

Suppose there are $K$ areas gathered in a set $I$, the time horizon being defined by initial and final times $\mathbb{T}_0$ and $\mathbb{T}_1$. The epidemiological state of an area is characterized by compartments, of $\mathcal{S}$usceptible, $\mathcal{E}$xposed, $\mathcal{I}$nfected, $\mathcal{Q}$uarantined, and $\mathcal{R}$ecovered individuals, considered as percentages of the total population in each area. The disease reproduction number is $R(t) \in [0, R_0]^K \subset \mathbb{R}^K$ while $T_{inc}$ and $T_{inf}$ are the incubation and infection periods. The coefficient

**Table 1. Model parameters.**

| Parameter | Definition | Value | Ref. |
|---|---|---|---|
| $\lambda$ | Rate of infected individuals who go to quarantine | | Eq (12) |
| $T_{inf}$ | Mean infectious time | 2.9 days | [38] |
| $T_{inc}$ | Mean incubation time | 5.2 days | [38] |
| $T_q$ | Mean quarantine time | $3\,T_{inf}$ | |
| $\sigma$ | % of SARS patients with COVID-19 | 25 | [39] |
| $\eta$ | Test efficiency | 0.8 | [40, 41] |
| $\tau$ | Time considered infectious before isolation | 1 to 3 days | |

$\alpha \in [0, 1]$ weighs the portion of the time that corresponds to the time spent outside their home city. Out of the 24h we consider that approximately 2/3 are spent on work and commuting. Since we simulate the equations using an integration method with a step of one day, we consider $\alpha = 1/3$, thus, we are taking a weighted proportion between working and night hours. For each area $i$, given an initial condition at time $\mathbb{T}_0$ the disease evolution during the night is described by the classical SEIQR model [28].

The mobility matrix, with size $|I| \times |I|$, has entries $p^{ki} \in [0, 1]$, representing the percentage of inhabitants of node $i$ traveling from $i$ to node $j$ [28]. Mobility data was provided by Brazilian company InLoco [42], obtained using high resolution smartphone geolocation. The proportion of people that did not leave the hub is accounted for in the diagonal of the mobility matrix. The graph of hubs can be classified as a small-world network, with a small-coefficient $\sigma = 1.8$ [43]. This means that although most hubs are not connected, all of them can be reached from every other by a small number of steps [44].

The impact of decreasing $R^i(t)$ on commuting is modeled as follows. For those nodes where $R^i(t)$ is below the natural reproduction number of the disease without intervention $R_0$, it is reasonable to assume that inbound travel will be discouraged. In our model we achieve this multiplying the respective entries of the mobility matrix by factor $R^i(t)/R_0$. This models the fact that if $R^i(t) = R_0$, then the city is not under any restriction and therefore normal travel can take place. On the other hand, to completely stop the spread of the virus it is also necessary to fully stop inbound travel. Letting $N^k$ denote the total population of node $k$, this discussion leads to the definition of the effective mobility matrix and the effective population, respectively defined by

$$\mathbb{p}^{ki}(t) := \frac{R^i(t)}{R_0} p^{ki} \quad \text{and} \quad \mathbb{N}^i(t) := \sum_{k \in I} \mathbb{p}^{ki}(t) N^k \quad \text{for } i \in I. \tag{4}$$

Commuting modifies the population circulation, yielding the infective

$$\mathbb{I}^j(t) := \frac{1}{\mathbb{N}^j(t)} \sum_{k \in I} \mathbb{p}^{kj}(t) \mathcal{I}^k(t) N^k \quad \text{for } j \in I. \tag{5}$$

Next, the following nonlinear function of the state variables

$$\mathcal{F}^i(t) = \frac{\alpha}{T_{\text{inf}}} R^i(t) \mathcal{S}^i(t) \mathcal{I}^i(t) + \frac{(1 - \alpha)}{T_{\text{inf}}} \sum_{j \in I} R^j(t) \mathbb{p}^{ij}(t) \mathcal{S}^i(t) \mathbb{I}^j(t) \tag{6}$$

describes the susceptible's dynamics, weighed in day and night portions.

Putting all these parameters together gives the following system of ordinary differential equations:

$$\dot{\mathcal{S}}^i(t) = -\mathcal{F}^i(t), \tag{7}$$

$$\dot{\mathcal{E}}^i(t) = \mathcal{F}^i(t) - \frac{1}{T_{\text{inc}}} \mathcal{E}^i(t), \tag{8}$$

$$\dot{\mathcal{I}}^i(t) = \frac{1}{T_{\text{inc}}} \mathcal{E}^i(t) - \frac{\lambda^i(t)}{T_{\text{inf}}} \mathcal{I}^i(t) - \frac{1}{T_{\text{inf}}} \mathcal{I}^i(t), \tag{9}$$

$$\dot{\mathcal{Q}}^i(t) = \frac{\lambda^i(t)}{T_{\text{inf}}} \mathcal{I}^i(t) - \frac{1}{T_{\text{q}}} \mathcal{Q}^i(t), \tag{10}$$

$$\dot{\mathcal{R}}^i(t) = \frac{1}{T_{\text{inf}}} \mathcal{I}^i(t) + \frac{1}{T_{\text{q}}} \mathcal{Q}^i(t). \tag{11}$$

The evolution depends on $\lambda^i(t)$, which determines the rate of individuals identified as

infectious when testing in the $i$th area at time $t$, supposing they are quarantined for $T_q$ days (we assume $T_q = 3T_{inf}$).

In practice, all patients with SARS need to be isolated and therefore go to the compartment $\mathcal{Q}$ while they wait for the test results. Notice however that, because of the initial conditions, at any given time $\mathcal{S} \gg \mathcal{I}$. As a result, the amount of susceptible individuals in the $\mathcal{Q}$ compartment is negligible when compared to the total susceptible population. Thus, we shall model the entry of infected individuals only in the $\mathcal{Q}$ compartment. In S5 File, we discuss a variant of the model where the quarantine is captured as the time delay system of equations leading to effective reproduction number. We show that the results are robust across both models.

### Effects of testing on λ

To determine $\lambda^i(t)$, we note that out of the $S_{sars}$ individuals with COVID-19 symptoms who seek attention in the health care system, only Cov individuals will be infected with COVID-19

$$\frac{\text{Cov}}{S_{sars}} = 1/4$$

as reported in Brazilian data [39]. As a result, if there are $\#T^i(t)$ tests performed in the $i$th area at time $t$ and $\tau$ is the time in days elapsed between becoming infectious and seeking assistance, then the fraction of detected individuals is

$$\lambda^i(t) = \eta e^{-\tau/T_{inf}} \frac{\text{Cov}}{S_{sars}} \frac{\#T^i(t)}{N^i I^i(t)}, \tag{12}$$

where $\eta = 0.8$ is the efficiency of the tests.

In our SEIQR model, the $\mathcal{I}$ compartment implicitly encompasses two subgroups representing symptomatic and asymptomatic individuals. Previous publications report that asymptomatic individuals in the $\mathcal{I}$ compartment are 2–4 times less likely to get tested than symptomatic individuals [11] but similar estimates are not available for Brazil. We adopted an alternative approach that is suitable to compare the performance of different testing protocols. Specifically, we implicitly account for asymptomatic by fixing a cap on the proportion of infected individuals that can be detected by testing. To define the cap, we consider the fraction $f_{symp}$ that corresponds to symptomatic individuals and nearby contacts in the infected compartment. Following [45], $f_{symp} = 40\%$ in our simulations. The cap is obtained by multiplying $f_{symp}$ by the number of tests that are necessary to identify a COVID-19 patient among people with SARS symptoms and, hence,

$$0 \leq \#T^i(t) \leq \frac{S_{sars}}{\text{Cov}} f_{symp} N^i \mathcal{I}^i(t). \tag{13}$$

Finally, we also consider the constraints $\sum_{i,t} \#T^i(t) \leq \#T_{tot}$ and $\#T^i(t) \leq \#T^i_{cap}$, which limit respectively the total number of tests employed in the campaign, and setting the daily cap for each area.

### Critical care beds occupancy

The solution procedure discretises the functional state $(\mathcal{S}, \mathcal{E}, \mathcal{I}, \mathcal{R}, \mathcal{Q})$ and control $(R, \#T)$ variables into vectors. For the infected individuals in particular, this means that $\mathcal{I}(t)$ is replaced by a vector with components $\mathcal{I}^i_t$ for $i \in I$ the set of areas, and $t \in \{\mathbb{T}_0, \mathbb{T}_0 + 1, \ldots, \mathbb{T}_1\}$, covering the days in the study ($\mathbb{T}_1 - \mathbb{T}_0 = 390$ days in our runs).

We set a probabilistic constraint for the use of ICU beds in each area, considering that the percentage of infected population that needs intensive care at time $t$ is uniform across the local

health area. Based on the structure of the considered uncertainty, the constraint can be cast into a deterministic equivalent reformulation. The procedure is explained in details in [28], we just mention a few key points here.

Bed usage is estimated from the ratio $\dfrac{v_t^i}{\sum\limits_{k=t-v}^{t} \mathcal{I}_k^i}$ , where the numerator, a known parameter, is the capacity constraint $v_t^i$ for the $i$th health area at time $t$. The denominator, an accumulation of sick individuals over $v$ days prior to $t$, represents the group among which a fraction may need critical care attention. Following [46], the average number of days infected individuals typically spend in the ICU is $v \in [7, 10]$ and we use $v = 7$ in our model.

The ratio stochastic process is then approximated by a time series, whose parameters are calibrated using historical records of intensive care unit beds and new infected individuals in the region. With our data, the best fit was an autoregressive model of lag 2. Consequently, the ratio is approximated by

$$\mathtt{icu}_t(\omega) = c_0 + c_1 t + \sum_{j=1}^{2} \phi_j \mathtt{icu}_{t-j} + \omega_t, \text{ where } \omega_t \sim \mathrm{iid}\, \mathcal{N}(0, \sigma_\omega^2). \tag{14}$$

In this expression, the white noise $\omega_t$ is a random variable that is independent and identically distributed according to a normal distribution with zero mean and variance given by $\sigma_\omega^2$. After calibration, the values for the parameters $c_0, c_1, \phi_1, \phi_2, \sigma_\omega$ are known and can be used to make explicit the following constraint

$$\mathbb{P}[\mathtt{icu}_t(\omega) \sum_{k=t-v}^{t} \mathcal{I}_k^i \leq v_t^i] \geq 0.90, \tag{15}$$

ensuring the local hospital capacity will not be exceeded, with 90% probability. The explicit deterministic equivalent formulation of this chance constraint is an affine constraint on $\mathcal{I}$, involving the inverse cumulative function of the standard Gaussian distribution, we refer to [28] for the complete development.

## Optimising on complex networks

Along the time horizon defined by the given initial and final times $\mathbb{T}_0$ and $\mathbb{T}_1$, having $K = 22$ administrative health districts under consideration, the epidemiological state of the region at time $t$ is characterised by the vector function

$$x(t) = (\mathcal{S}, \mathcal{E}, \mathcal{I}, \mathcal{Q}, \mathcal{R})(t) \in [0, 1]^{5 \times K}, \text{ for } t \in [\mathbb{T}_0, \mathbb{T}_1]. \tag{16}$$

As mentioned, the discretisation of the SEIQR system of differential equations is performed using a central finite differences scheme; see [28]. For Sao Paulo state specifically, this amounts to approximate the state function $x(t)$ by a large-scale vector; for instance the component $\mathcal{S}(t)$ is replaced by $\mathcal{S}_t^i$ for $i \in I = \{1, \ldots, K = 22\}$ and $t \in \{1, \ldots, 390\}$, and similarly for the control $R^i(t)$ and other variables, including the number of tests performed in each area, $\#T^i(t)$. Accordingly, given the initial state $x_0$ and the basal reproduction number $R_0 = 1.8$, the mathematical

optimization problem to be solved is a large-scale nonlinear program in the form

$$
\begin{cases}
\min\limits_{\substack{R_t^i \in [0, R_0] \\ \#T \geq 0}} \quad \sum\limits_{t=1}^{390} \psi_t(R_t) \\[2ex]
\text{s.t.} \quad (x, R, \#T) \quad \text{satisfies the discretized SEIQR in Methods} \\[1ex]
\phantom{\text{s.t.} \quad} (x, \#T) \quad \text{satisfies constraints on testing and ICU beds}.
\end{cases}
\tag{17}
$$

The objective function $\psi_t$ depends on $R_t = (R_t^1, \ldots, R_t^K)$, and it can assess the control performance combining several terms. In our simulations we used the simple expression $\psi_t^{MaxCirc}(R_t) = \sum\limits_{i \in I} w^i(R_0 - R_t^i)$ that ensures maximal circulation for given weights $w^i$ proportional to the areas' population. Finally, following the public policy in Sao Paulo state, changes in the controls $R_t^i$ of each city are only possible every 2 weeks.

The different test distribution protocols in Fig 2 were determined by applying the optimization framework as follows. Solving (17) provides for every day, in every region, the optimal value for the control and state variables. The latter yield the dynamic of the pandemia that is observed as a result of the control variables. With Smart Testing, there are two types of control variables: the number of tests to be applied and the corresponding levels of the reproduction number (reproduction numbers have associated different levels of measures restricting circulation). By contrast, with the on-demand strategy the number of tests is no longer a variable in (17), but a parameter. This is achieved simply by fixing in (17) the value of the number of tests to be proportional to the population of the region. For the hub testing, the number of tests in the smaller regions is set to zero, and it is let as a variable in the hubs.

## Supporting information

**S1 File. Flowchart.**
(PDF)

**S2 File. Model calibration.**
(PDF)

**S3 File. New York City.**
(PDF)

**S4 File. Additional scenarios in Sao Paulo state.**
(PDF)

**S5 File. Effective models.**
(PDF)

**S1 Video.**
(MP4)

**S2 Video.**
(MP4)

## Acknowledgments

## Code availability

Input files or sets of input parameters as well as the codes are available in https://github.com/pjssilva/Robot-dance.

## Author Contributions

**Conceptualization:** Paulo J. S. Silva, Tiago Pereira, Claudio J. Struchiner.

**Formal analysis:** Claudia Sagastizábal.

**Funding acquisition:** Tiago Pereira.

**Investigation:** Paulo J. S. Silva, Tiago Pereira.

**Methodology:** Paulo J. S. Silva, Tiago Pereira, Claudia Sagastizábal.

**Resources:** Paulo J. S. Silva.

**Software:** Paulo J. S. Silva, Claudia Sagastizábal, Marcelo M. Cordova.

**Validation:** Marcelo M. Cordova.

**Visualization:** Luis Nonato, Marcelo M. Cordova.

**Writing – original draft:** Tiago Pereira, Claudia Sagastizábal.

**Writing – review & editing:** Paulo J. S. Silva, Tiago Pereira, Claudia Sagastizábal, Claudio J. Struchiner.

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
