## [Decision Letter · Decision Letter 0]

26 Mar 2021

PONE-D-21-06401

Smart testing and critical care bed sharing for COVID-19 control

PLOS ONE

Dear Dr. Pereira,

Thank you for submitting your manuscript to PLOS ONE. After careful consideration, we feel that it has merit but does not fully meet PLOS ONE’s publication criteria as it currently stands. Therefore, we invite you to submit a revised version of the manuscript that addresses the points raised during the review process.

We look forward to receiving your revised manuscript.

Kind regards,

Martial L Ndeffo Mbah, Ph.D

Academic Editor

PLOS ONE

Additional Editor Comments:

The main issues to be addressed are the model formulation and providing supporting references to some of the statements and value referred to as being informed by data. The model explicitly account for asymptomatic infection. One of the key parameter to the analysis is the rate at which people are seeking for care, but it is based on data from people with covid-19 like symptoms therefore it does not account for asymptomatic who are less likely to seek care and be tested and isolated. Those asymptomatic individuals will spread the disease for a longer time period that tested symptomatic individuals. By not modeling asymptomatic explicitly, the model is likely to overestimate the effectiveness of the intervention. A Us-based study has shown that symptomatic individuals are 2 to 4 times more likely to be tested for covid-19 than asymptomatic https://www.nature.com/articles/s41562-020-00944-2#Sec4

In addition these comments, please thoroughly address the issues raised by the three reviewers.

Journal Requirements:

4. We note that Figures 1-3 and Supporting Information files 'n.pdf', 'icu_panel.mp4' and 'ntests_relpop_panel.mp4'  in your submission contain map images which may be copyrighted. All PLOS content is published under the Creative Commons Attribution License (CC BY 4.0), which means that the manuscript, images, and Supporting Information files will be freely available online, and any third party is permitted to access, download, copy, distribute, and use these materials in any way, even commercially, with proper attribution. For these reasons, we cannot publish previously copyrighted maps or satellite images created using proprietary data, such as Google software (Google Maps, Street View, and Earth). For more information, see our copyright guidelines: http://journals.plos.org/plosone/s/licenses-and-copyright.

4.1.    You may seek permission from the original copyright holder of Figures 1-3 and Supporting Information files 'n.pdf', 'icu_panel.mp4' and 'ntests_relpop_panel.mp4' to publish the content specifically under the CC BY 4.0 license. 

4.2.    If you are unable to obtain permission from the original copyright holder to publish these figures under the CC BY 4.0 license or if the copyright holder’s requirements are incompatible with the CC BY 4.0 license, please either i) remove the figure or ii) supply a replacement figure that complies with the CC BY 4.0 license. Please check copyright information on all replacement figures and update the figure caption with source information. If applicable, please specify in the figure caption text when a figure is similar but not identical to the original image and is therefore for illustrative purposes only.

5. We note you have included a table to which you do not refer in the text of your manuscript. Please ensure that you refer to Table 'Table with Parameters' in your text; if accepted, production will need this reference to link the reader to the Table.

Reviewers' comments:

Reviewer's Responses to Questions

**Comments to the Author**

1. Is the manuscript technically sound, and do the data support the conclusions?

Reviewer #1: No

Reviewer #2: Yes

Reviewer #3: Partly

2. Has the statistical analysis been performed appropriately and rigorously? 

Reviewer #1: N/A

Reviewer #2: Yes

Reviewer #3: Yes

3. Have the authors made all data underlying the findings in their manuscript fully available?

Reviewer #1: Yes

Reviewer #2: Yes

Reviewer #3: Yes

4. Is the manuscript presented in an intelligible fashion and written in standard English?

Reviewer #1: Yes

Reviewer #2: Yes

Reviewer #3: Yes

5. Review Comments to the Author

Reviewer #1: The article "Smart testing and critical care bed sharing for COVID-19 control" by Pereira et al. investigates the optimal use of PCR test allocation and ICU bed sharing to mitigate COVID-19 transmission in the Sao Paulo region of Brazil using a compartmental model of COVID-19 transmission embedded within an optimal control framework. The general methodology and derived results are interesting, however I have serious concerns about the structure of the underlying transmission model. In particular, the model does not seem to capture either asymptomatic or presymptomatic transmission of infected individuals - both of which are understood to contribute substantially to overall transmission. (Whilst in the Methods section a claim is made that the latter is accounted for through the parameter \\zeta, I found this claim unconvincing and believe that more natural ways of including this effect exist). Therefore, I believe the base model is unsuitable for the desired application and, as such, I must recommend that this article not be accepted for publication.

In addition to the above, please find further comments below:

- Throughout the text several claims are made without reference

- Why did the authors choose to use the R0 value for the second wave in Germany? Surely there is sufficient data from Brazilian cases to estimate the local reproduction number?

- Lines 121 - 129: This paragraph is repeated from above

- The discussion contains no reflection on the limitations of the modelling framework and analysis

- I believe that the interpretation of \\lambda as the percentage of infected individuals that are quarantined is incorrect, from equation (10) it appears that the true percentage is \\lambda / (1 + \\lambda).

- In general, I found the methods section difficult to follow with several key parameters left undefined

Reviewer #2: The present manuscript analyses the problem of optimising ICU bed occupancy in the current COVID-19 pandemics and the effect of different testing strategies.

The paper is nice, relevant and elegant although some details could be the text. I reckon the paper can be useful as case-study even for other stances such as ICU bed sharing in the UE.

Nonetheless, I would like to see some points further discussed.

- The authors mention their study spans the period between 1st July 2020 and 31st July 2021, but today is the 23rd March 2021. is it between 1st July 2019 and 31st July 2020 instead?

- The epidemiological model used in the paper is similar to that used in the following paper for Brazil as well

https://www.worldscientific.com/doi/abs/10.1142/S0129183120501351

- The impact of testing strategies is largely impacted by the so-called asymptomatic cases, who are not prompt to go testing. Models considering such state (eg https://www.nature.com/articles/s41598-020-76257-1 and https://arxiv.org/abs/2005.09019)

- A mobility model should discuss the economical problem of people who cannot isolate because they need to get income.

- Further details on the network structure used in the work should be provided. Do authors consider small world, scale-free, multi-layer networks?

Reviewer #3: The authors have presented a very interesting study, combining a rigorous mathematical modelling approach with effective use of data and a discussion of real-world clinical consequences. A novel optimisation approach is used, in which the authors calculate the maximum amount of epidemic growth allowed given limits on specific clinical outcomes; this allows them to propose testing approaches which are effective in reducing morbidity and mortality but also allow for a maximum of "normal" social behaviour.

My main critique of the manuscript is that the smart testing methodology is not sufficiently explained within the main paper. The application of smart testing is the main focus of the paper, but the description of smart testing is limited to a short outline at the start of the results section on Page 4. In particular, this very general account makes it difficult to know exactly how smart testing differs from the on-demand and hub testing strategies. For publication, the authors need to add a few paragraphs to the methods section of the manuscript explaining the principles behind their smart testing approach, and highlighting the contrasts with the other two strategies.

My other criticisms are fairly minor. Throughout the manuscript, the authors use a lower-case "r" for reproductive ratios. Typically in infectious disease modelling, lower case "r" is reserved for growth rates, with upper case "R" used for reproductive ratios. To improve the clarity of their paper, the authors should bring their notation into line with this convention. Also throughout the paper, the authors hyphenate the phrase "social distancing", writing it as "social-distancing". There is no need for a hyphen here, and a hyphen is not used in, for instance, UK or USA government guidelines on social distancing, and so the authors should replace these hyphens with spaces.

In the abstract, the authors talk about "The economic infrastructure of the country" when talking in general about developing countries. Because they are talking about a group of countries, this should say "The economic infrastructure of these countries".

Bottom of page 6, line 113: “The model assumes the basic reproductive number with sanitary measures is r0 = 1.8 (the value observed for the second wave in Germany [11]).” The basic reproductive ratio is highly dependent on the social setting, and given the socioeconomic contrasts between Germany and Brazil (which the authors do an excellent job of highlighting), the value in Germany may not be a useful indicator of the value in Brazil. It is also worth pointing out that reference 11 points to a website where an R0 estimate is not immediately visible! If possible, the authors should find an estimate specifically from Brazil, or if this is not possible try to find a range of R0 values across different socioeconomic settings, and compare their chosen value with this range.

From page 6 line 113 to page 7 line 129 we get two phrasings of the same paragraph. This appears to be a proofreading error, and one of these paragraphs should be deleted.

Page 7, line 134: “When ri_no test(t) = r0, the basal reproduction number of the new normal, there are no travel restrictions.” It is unclear to me what the phrase “new normal” refers to – here the authors appear to mean the situation in which there are no restrictions, but “the new normal” is popularly used to mean life with social distancing measures imposed. I suggest using a more precise description like “the baseline reproduction number in the absence of interventions”. The causality in this sentence is also the wrong way round: ri_no test = r0 because of the absence of travel restrictions, whereas the authors make it sound like the value of the reproductive ratio is what causes the absence of travel restrictions. The authors should make it clear that this parameter value reflects, rather than determines, reality.

Page 15, line 270-271: “The coefficient α(t) ∈ [0; 1] weighs the portion of the considered time t that corresponds to the time spent outside their home city, so α(t) = 1/3”. I can not see anything suggesting that the value of this coefficient is 1/3, and I am not sure if the authors mean something more like “we will set α(t)=1/3”. For publication, the authors should explain whether this value is estimated from some data or is chosen arbitrarily; given the challenges surrounding mobility data this assumption seems reasonable.

On page 17, there is a section called “Table with parameters”, containing no text. I think this is meant to contain the parameter table but because of the way LaTex formats tables the table appears on the next page. The authors should delete this section heading, moving the table into another subsection and referencing it in the main body text.

6. PLOS authors have the option to publish the peer review history of their article (what does this mean?). If published, this will include your full peer review and any attached files.

Reviewer #1: No

Reviewer #2: No

Reviewer #3: **Yes: **Joe Hilton

---

## [Author Response · Author response to Decision Letter 0]

21 May 2021

Reply to the referee comments are attached as Response to Reviewers file

---

## [Decision Letter · Decision Letter 1]

16 Jun 2021

PONE-D-21-06401R1

Smart testing and critical care bed sharing for COVID-19 control

PLOS ONE

Dear Dr. Pereira,

Thank you for submitting your manuscript to PLOS ONE. After careful consideration, we feel that it has merit but does not fully meet PLOS ONE’s publication criteria as it currently stands. Therefore, we invite you to submit a revised version of the manuscript that addresses the points raised during the review process.

We look forward to receiving your revised manuscript.

Kind regards,

Martial L Ndeffo Mbah, Ph.D

Academic Editor

PLOS ONE

Journal Requirements:

Additional Editor Comments (if provided):

Thoroughly address the reviewers comments as they will greatly improve the quality and readability of the manuscript

Reviewers' comments:

Reviewer's Responses to Questions

**Comments to the Author**

1. If the authors have adequately addressed your comments raised in a previous round of review and you feel that this manuscript is now acceptable for publication, you may indicate that here to bypass the “Comments to the Author” section, enter your conflict of interest statement in the “Confidential to Editor” section, and submit your "Accept" recommendation.

Reviewer #1: All comments have been addressed

Reviewer #3: (No Response)

2. Is the manuscript technically sound, and do the data support the conclusions?

Reviewer #1: Yes

Reviewer #3: Yes

3. Has the statistical analysis been performed appropriately and rigorously? 

Reviewer #1: Yes

Reviewer #3: Yes

4. Have the authors made all data underlying the findings in their manuscript fully available?

Reviewer #1: Yes

Reviewer #3: Yes

5. Is the manuscript presented in an intelligible fashion and written in standard English?

Reviewer #1: Yes

Reviewer #3: (No Response)

6. Review Comments to the Author

Reviewer #1: See attachment.

Reviewer #3: I thank the authors for their response to the comments of myself and the other reviewers. Unfortunately I do not feel that the authors have sufficiently addressed all of my comments. In particular, the paper still lacks an explicit explanation of what their smart testing regime actually involves, and a clear definition and justification for the parameter alpha. I would also like to point out that the line references in the authors’ response were incorrect, and I hope that in any further rounds of review they are careful to make sure this does not happen again.

Detailed notes:

I am still not satisfied that the authors have given an explicit definition of the smart testing regimen. My understanding is that by smart testing, the authors mean solving an optimal control problem and conducting tests according to the solution to that optimal control problem. The newly added paragraph on page 22 talks a bit about the optimal control problem, but the paper as a whole still lacks any explicit statement that smart testing means solving that optimal control problem and conducting tests accordingly. The authors need to provide a definition of smart testing in the introduction, and ideally in the abstract. This definition does not need to involve any mathematical detail, but should let the reader know unambiguously that the smart testing regimen means distributing tests according to where the optimal control problem suggests they will have the largest impact.

At the bottom of page 7 and going over to page 8, the authors says that “when R^i_(no test)(t)=R_0, no travel restrictions are necessary”. While, having read the paper a few times, I understand what the authors mean here, on first reading this statement is extremely counterintuitive. The basic reproductive ratio is the reproductive ratio of cases in a population undergoing “normal” behaviour with no population-level immunity. This means we almost always assume that the effective reproductive ratio will be smaller than the basic reproductive ratio, partly because NPIs seek to reduce social mixing, but also because even in the absence of NPIs the accumulation of population-level immunity will reduce the stock of susceptibles available to each case. This means that to get an effective reproductive ratio higher than R_0, we need either an ongoing intensification of social contacts, or rapid population growth exceeding the rate of accumulation of immunity. Of course, on reflection it is clear that since R^i(no test)(t) is a maximum value for effective reproductive ratio given some restraints, what the authors are saying is that if we can keep hospitalisations below 90% of capacity with an effective reproductive ratio equal to R_0, then we don’t need to implement restrictions. I recommend that the authors add a statement to this effect, making it very explicit that their aim is to find the maximal reproductive ratio which keeps hospital occupancy below 90% of capacity. While I am aware I have already requested on notation change in this context, some confusion could also be avoided by using slightly different notation to avoid confusion between this maximum and the actual value of the effective reproductive ratio – for instance by adding a hat or tilde to R^i_(no_test)(t). The authers subsequently define R^i_test(t) to be the instantaneous observed reproductive ratio (not a maximum!) under a given testing regime, which amplifies this confusion and the need to explicitly distinguish between maxima and attained values.

Unfortunately, I am not convinced that the authors have sufficiently addressed my query regarding the parameter alpha, the proportion of time per day spent outside of an individual’s resident location. The authors say “Out of the 24h we consider that approximately 2/3 are spent on work and commuting. Since we simulate the equations using an integration method with a step of one day, we consider alpha = 1/3, thus, we are taking a weighted proportion between working and night hours.” It is extremely unclear to me what this means. The first sentence appears to mean that 16 hours per day are spent outside of one’s resident location; while the authors do not supply any reference to data, this seems like an overestimate. While I can imagine that in very economically deprived contexts individuals may indeed need to work away from home for 16 hours at a time, this seems very high for the time per day that an average individual spends away from their home location on an average day. This would also give alpha=2/3, based on the definition they give of alpha as the proportional amount of time spent outside of the household, rather than alpha=1/3. I am not sure what the authors mean by “a weighted proportion between working and night hours”; if they are treating “night” and “home” contacts as the same thing, I would recommend sticking to “home” as the terminology here. I note also that in Equation 6 on page 18 they appear to use alpha as the coefficient for within-patch mixing, in which case (by their assumption about commute/work time), alpha would indeed by 2/3. Again, in the description of Equation 6 they talk about “day and night” portions, and they either need to replace this with “work and home” or else state very clearly what their assumptions are about how work and home contacts are distributed over day and night. The authors need to check that their written definition of alpha actually corresponds to its role in the model, accompanied with a simple explanation of their choice of value along the lines of “we choose a value of alpha equal to 1/3 (or 2/3) on the assumption that on average individual spends 8 (or 16) hours per day away from their home location”.

Spelling/grammar/style notes:

Page 2: “In this situation, nonlinear effects resulting from the high degree of population mobility makes the decision of optimal test distribution a real challenge also because many individuals cannot isolate as they need to get income” – I realise the end of this sentence was added in response to another reviewer’s comments, but I am afraid the wording here is rather confusing, and the note about economic factors may be better off in its own sentence.

Page 5: Incorrect date of “July 31, 2021” is still there!

Caption to Figure 1 on page 7: “Thicker edges represent larger commuting.” – this should say something like “Thicker edges represent more intensive commuting.”

Page 11: The notation τ appears to be used before it is defined.

Figure 3 on page 13 still includes the phrase “new normal”, despite its removal from the rest of the paper.

Newly added paragraph on page 22 makes repeated reference to Equation 18; there is no Equation 18 in the paper and I think the authors mean to be referencing Equation 17 here.

7. PLOS authors have the option to publish the peer review history of their article (what does this mean?). If published, this will include your full peer review and any attached files.

Reviewer #1: **Yes: **Michael T. Meehan

Reviewer #3: No

---

## [Author Response · Author response to Decision Letter 1]

26 Jul 2021

Response to reviewers question are in the File: Response to Reviewers.pdf

---

## [Decision Letter · Decision Letter 2]

23 Aug 2021

PONE-D-21-06401R2

Smart testing and critical care bed sharing for COVID-19 control

PLOS ONE

Dear Dr. Pereira,

Thank you for submitting your manuscript to PLOS ONE. After careful consideration, we feel that it has merit but does not fully meet PLOS ONE’s publication criteria as it currently stands. Therefore, we invite you to submit a revised version of the manuscript that addresses the points raised during the review process.

We look forward to receiving your revised manuscript.

Kind regards,

Martial L Ndeffo Mbah, Ph.D

Academic Editor

PLOS ONE

Journal Requirements:

Additional Editor Comments (if provided):

There are two very minor issues (typos) that need to be addressed. See reviewer #3

Reviewers' comments:

Reviewer's Responses to Questions

**Comments to the Author**

1. If the authors have adequately addressed your comments raised in a previous round of review and you feel that this manuscript is now acceptable for publication, you may indicate that here to bypass the “Comments to the Author” section, enter your conflict of interest statement in the “Confidential to Editor” section, and submit your "Accept" recommendation.

Reviewer #1: All comments have been addressed

Reviewer #3: All comments have been addressed

2. Is the manuscript technically sound, and do the data support the conclusions?

Reviewer #1: Yes

Reviewer #3: Yes

3. Has the statistical analysis been performed appropriately and rigorously? 

Reviewer #1: Yes

Reviewer #3: Yes

4. Have the authors made all data underlying the findings in their manuscript fully available?

Reviewer #1: Yes

Reviewer #3: Yes

5. Is the manuscript presented in an intelligible fashion and written in standard English?

Reviewer #1: Yes

Reviewer #3: Yes

6. Review Comments to the Author

Reviewer #1: I am satisfied that all comments have been addressed.

Reviewer #3: I am pleased to say that the authors have addressed all the points from my previous set of comments, although the newly-added definition of the parameter tau needs to be revised for grammar and clarity. The complete sentence currently reads as follows: “The results reported in Figure 3 were obtained under the premise that susceptible individuals seek healthcare assistance and are tested by RT-PCR where τ = 1 day after experiencing symptoms, where τ the number of days it takes for an infectious to seek assistance”. I think this sentence really needs to be split in two, along the lines of “We define τ to be the number of days it takes for an infectious individual to seek assistance. The results reported in Figure 3 were obtained under the premise that susceptible individuals seek healthcare assistance and are tested by RT-PCR on average one day after experiencing symptoms, so that τ = 1”.

My only other comment (and I apologise for not catching this on the first draft) is that the authors switch between using SEIQR and SEIRQ to denote their compartmental structure in their methods section. This should be corrected so that one acronym is used consistently.

Pending these two corrections, the manuscript is entirely suitable for publication.

7. PLOS authors have the option to publish the peer review history of their article (what does this mean?). If published, this will include your full peer review and any attached files.

Reviewer #1: No

Reviewer #3: **Yes: **Joe Hilton

---

## [Author Response · Author response to Decision Letter 2]

25 Aug 2021

Response to reviewers question are in the File: Rebuttal_2021.08.24.pdf

---

## [Editor Report · Decision Letter 3]

27 Aug 2021

Smart testing and critical care bed sharing for COVID-19 control

PONE-D-21-06401R3

Dear Dr. Pereira,

We’re pleased to inform you that your manuscript has been judged scientifically suitable for publication and will be formally accepted for publication once it meets all outstanding technical requirements.

Kind regards,

Martial L Ndeffo Mbah, Ph.D

Academic Editor

PLOS ONE
---

## [Editor Report · Acceptance letter]

16 Sep 2021

PONE-D-21-06401R3 

Smart testing and critical care bed sharing for COVID-19 control 

Dear Dr. Pereira:

I'm pleased to inform you that your manuscript has been deemed suitable for publication in PLOS ONE. Congratulations! Your manuscript is now with our production department. 

Kind regards, 

on behalf of

Dr. Martial L Ndeffo Mbah 

Academic Editor

PLOS ONE